# A New Perspective on Supporting Vulnerable Road Users’ Safety, Security and Comfort through Personalized Route Planning

**DOI:** 10.3390/ijerph20043027

**Published:** 2023-02-09

**Authors:** Diogo Abrantes, Marta Campos Ferreira, Paulo Dias Costa, Joana Hora, Soraia Felício, Teresa Galvão Dias, Miguel Coimbra

**Affiliations:** 1Faculdade de Ciências, Universidade do Porto, 4169-007 Porto, Portugal; 2Faculdade de Engenharia, Universidade do Porto, 4200-265 Porto, Portugal; 3INESC TEC—Instituto de Engenharia de Sistemas e Computadores, Tecnologia e Ciência, 4200-465 Porto, Portugal; 4Faculdade de Medicina, Universidade do Porto, 4200-319 Porto, Portugal; 5CINTESIS—Centro de Investigação em Tecnologias e Serviços de Saúde, 4200-450 Porto, Portugal

**Keywords:** route planner, active travel, sustainable mobility, users’ perceptions, customization, IoT

## Abstract

Due to an increase in population, urban centers are currently seeing an increase in traffic, resulting in negative consequences such as pollution and congestion. Efforts have been made to promote a modal shift towards the use of more sustainable means of transport, such as walking and cycling, but several deterrents influence the citizens’ perceptions of safety, security and comfort, discouraging their choice of active modes of transport. This study focuses on the importance of providing meaningful information to vulnerable road users (VRUs) to support their perceptions and objectives while moving within urban spaces through a novel concept of route planning. A broad survey of the needs and concerns of VRUs through interviews, focus groups and questionnaires, applied to the Portuguese population of the Metropolitan Area of Porto, led to the development of a new concept of route planners that show personalized routes according to the individual perceptions of each user. This concept is materialized in a route planner prototype that has been extensively tested by potential users. Subjective evaluation and feedback showed the usefulness of the concept and added value to a familiar product, leading to a satisfying experience for participants. This study shows that there is an opportunity to improve these tools to provide a higher degree of power and customization to users on route planning, which includes addressing mobility restrictions and personal perceptions of safety, security and comfort. The ultimate goal of this new approach is to persuade citizens to switch to more sustainable means of transport.

## 1. Introduction

Cities shelter 56% of the world’s current population, and with a predicted increase to 68% by 2050, this means that about 2.5 billion people could be added to urban areas by then [1,2], bringing various environmental and social challenges. A population increase means more people are on the move, and citizens adopt private vehicles as the main means of transport to satisfy those needs [3]. City centers experience an increase in traffic, which in turn brings associated harmful effects. To counter this, efforts have been made to promote a modal shift to the use of more sustainable transport means, such as walking, the use of bicycles and scooters, as well as traditional public transport [4,5,6,7,8]. Recently, a number of studies that measured a reduction of air pollutants from vehicles in city centers due to the pandemic lockdown were published [9,10,11,12], which further reinforces the point.

The WHO European Healthy Cities program has been adopting a series of objectives for the Healthy Urban Planning initiative. Among them, the promotion of healthy lifestyles and regular exercise, promoting safety and a sense of security, as well as providing attractive environments with acceptable noise levels and good air quality were part of the main themes for cities to develop within urban planning [13]. Leaning on that, agencies determine an overall air pollution category and issue recommendations for the general public and city policymakers [14], which are of paramount importance due to the evidence linking increased air pollution to respiratory and cardiovascular diseases [15,16]. Moreover, continuous exposure to urban noise levels is also a serious and underestimated issue [17], being related to increased stress caused to humans and animals, and linked to discomfort, sleep and cognitive performance in adults and children [18].

Fortunately, personal health and well-being concerns are on the rise, particularly in urban areas. Increased longevity, adherence to exercising in public spaces, the growing support for citizens with reduced mobility in urban settings, the popularization of healthier nutritional choices, and the use of sensors that monitor health parameters [19] all help promote more sustainable transport means.

The European Union’s Intelligent Transport Systems (ITS) Directive defines Vulnerable Road Users (VRUs) as “non-motorised road users, such as pedestrians and cyclists as well as motorcyclists and persons with disabilities or reduced mobility and orientation” [20]. There are several factors affecting the level of safety of VRUs. The number of road accidents and fatalities in urban areas with pedestrians and cyclists [21], weather conditions, lighting levels, a poor or lack of maintenance of infrastructures [22] ultimately influence the citizen’s safety and security perceptions [23], discouraging the choice of active modes of transport [24]. VRUs claim they feel more likely to be targeted by criminals [25], and fewer women than men declare feeling safe while being more conscious of the presence (or lack thereof) of security monitoring systems and personnel [26,27]. Other factors that influence decision-making are related to traveling time, price, parking, terrain type and slope, and neighborhood or environment aspects [28,29].

As technological and digital innovation are crucial for the development of improvements in the public administration, mobility, environment, economy, and quality of life in cities [30], we can convey that smart cities bring advantages for modern societies’ quality of life, such as increased free time and security, energy efficiency, and pollution reduction [31]. As smart cities are shaped by intelligent devices and sensor networks, rising challenges are usually tackled with the introduction of new technologies, especially those associated with the Internet of Things (IoT). IoT substructures can simplify operations and affect the different features of a citizen’s life by creating cost-effective municipal services, enhancing public transformation and reducing traffic congestion while keeping them safer and healthier [32]. To this end, they rely on sensors to gather specific data regarding public transport information, traffic status, weather conditions, lighting levels, air and noise pollution status and energy consumption, among others, while other intelligent devices can feed surveillance and monitoring systems.

This study focuses on the importance of providing meaningful information to the VRUs to support their perceptions of personal safety, security and comfort while moving within urban spaces. This is achieved through a new approach to route planning that combines the potential of the IoT and users’ personal perceptions and objectives.

Route planners are widely used and focus on helping users find the best path between two points while considering their mobility preferences. Current options mostly focus on two dimensions, namely time and distance, which might not be enough to support other personal perceptions. Considering new aspects and different dimensions, such as air quality, noise levels, accessibility, or people density, all of which are sustained by recent advances in the area of sensorization and the IoT [33], it is possible to envision a tool to persuade the users’ shift to more sustainable means of transport.

People are also different. Catering to different moods or desires is interesting, but catering to specific needs has a different weight to it. Even on foot, it is necessary to consider mobility limitations affecting people that might not be accounted for, such as being pregnant, the use of a baby stroller, having a health condition, being older, or having a disability.

Previous works concerning active mobility and overall commuting have shown the importance of a subjective well-being parameter associated with a route or commute, which is affected by factors such as duration, crowding, or unpredictability [34]. Other factors such as geographical (distance, slope) or environmental data (air quality, noise levels) are also already being looked at as crucial when proposing routes that aim to minimize stress exposure to active mobility users [35,36,37].

The next section presents the new approach to route planners for VRUs proposed in this article, which is materialized in a prototype detailed in Section 3. Section 4 discusses the main results and Section 5 presents the conclusions.

## 2. A New Approach on Route Planners for VRUs

To determine how the proposal could be implemented and adopted, a set of User-Centered Design (UCD) methods focused on evaluating user needs were employed. The concepts of usability and user experience (UX) were also integrated into the evaluation of the prototype to ensure that the technology and services being developed were accessible to all users. The methodological approach used to conceptualize the proposed route planner is illustrated in Figure 1. The following section outlines the methodological approach taken and the main results obtained. The proposed concept is then introduced.

### 2.1. Methodological Approach and Results

The methodology employed in the development of the idea focused on understanding potential user needs. The process included the following steps, the methods and results of which will be described in the subsequent sections:An online survey to the general audience to gather socio-demographic information, common impressions, preferences, and habits;Focus group and interview sessions to collect information about global and individual perceptions of safety, security and comfort when moving, considering personal characteristics and specific circumstances of travel;A streamlined comparative analysis of currently related apps to gauge a few strengths and weaknesses.

The combination of these steps enabled a thorough understanding of the different perspectives of citizens regarding their safety, security, and comfort while using active modes of transportation within the city. This understanding formed the basis for the development of the innovative concept of a route planner proposed in this article.

#### 2.1.1. Online Survey

A three-part online survey was posed to characterize the studied sample, including questions about general data and socio-demographic characterization; attitudes towards information and communication technologies in general and not mobility-wise, using questions adapted from [38], since the goal of this study was the proposal of a technology-based solution to which the sample should be open to; and general impressions about the area in question, along with a few preferences, habits, wants and needs. This was meant to understand possible usage or acceptance rates, as well as opportunities to address. The target population was selected using a convenience sampling technique and included all individuals that were at least 18 years old at the time the survey was deployed while agreeing to participate in the study. To ensure data accuracy, respondents that had the survey opened for less than 60 s, for more than 2000 s, or had an overall response rate of less than 33%, were excluded from the study.

In order to achieve a broader sample of people with special needs, such as mobility issues or other impairments, could prove difficult, the questionnaire was also sent to the Portuguese Foundation for Cardiology, the Portuguese Association for the Disabled, the Portuguese Association for the Blind and Amblyopes, as well as the Portuguese Association for Asthmatics, in hopes of receiving some feedback and/or participants for the upcoming focus groups.

This study is based on a prospective and observational methodology during a 3-month period. The Metropolitan Area of Porto (AMP) is 1,736,228 inhabitants and approximately 80% (1,388,982) of that population is mobile. The demographic data for the population of the AMP in 2021 indicates a 47.4% male and 52.6% female split. The age distribution is as follows: 10.9% are between the ages of 20–29, 12.0% are between 30–39, 15.2% are between 40–49, 15.2% are between 50–59, and 29.1% are over 60 years of age [39].

The questionnaire had 326 total respondents, with 84 being excluded for not matching the selection criteria, resulting in a sample size de 242 participants. It can be noted that this sample size is in line with the size of the samples used in related works, such as [23,40,41], and can lead to statistically significant results assuming a confidence level of 85% considering the aggregated sample, as the minimum sample size is 208 individuals for such as confidence level [42]. The studied sample is a convenience sample and not a stratified sample, in which all participants answered all questions.

Through a descriptive statistical analysis, it is possible to verify that 137 (57.1%) of the participants were female, and the most frequent age range was from 40 to 49 (65; 27%). Regarding formal education, 198 participants (83.5%) had an ISCED 6 level (Bachelor’s or equivalent). Most participants (219; 93.6%) had no mobility issues, with 15 (6.4%) experiencing some type of limitation, such as cardiac or respiratory disease, physical or visual impairment and being pregnant. These findings are summarized in Table 1. Regarding the use of the area defined as Porto’s city center, most participants used it either for leisure (143; 21.5%), commerce & services (140; 21.1%) and/or culture & events (127; 19.1%). Preferred means of transport to access the area include car (131; 33.4%) and subway (93; 23.7%), with 69 participants (17.6%) traveling on foot. The most commonly identified obstacles to mobility were irregular or damaged pavement (136; 23.4%), vehicles parked in pedestrian zones (121; 20.8%) and insufficient parking spaces (112; 19.3%).

Concerning attitudes towards technology, questions 1.1 through 1.9, on question 1.1 (I feel it is important to be able to find any information whenever I want online), 97 (45.8%) and 89 (42.0%) participants, respectively, agreed or strongly agreed with the statement. On question 1.2 (I feel it is important to be able to access the Internet any time I want), 71 (33.3%) and 75 (35.2%) participants, respectively, disagreed or neither agreed nor disagreed with the statement. As for question 1.3 (I think it is important to keep up with the latest trends in technology), most participants agreed (90; 42.3%) or strongly agreed (109; 51.2%) with the statement. In question 1.4 (Technology will provide solutions to many of our problems), 67 (32.4%) and 93 (44.9%) participants neither agreed nor disagreed, or agreed, respectively, with the statement. In question 1.5 (With technology, anything is possible), a vast majority of participants either agreed (130; 61.0%) or strongly agreed (55; 25.8%) with this statement. Regarding question 1.6 (I feel that I get more accomplished because of technology), 70 (33.0%) and 74 (34.9%) of participants disagreed or neither agreed nor disagreed, respectively. For question 1.7 (New technology makes people waste too much time), 60 (28.3%) and 72 (34.0%) participants, respectively, either disagreed or neither agreed nor disagreed with the statement. In question 1.8 (New technology makes life more complicated), results showed that most participants either agreed (71; 33.3%) or disagreed (58; 27.2%) with this statement. Finally, in question 1.9 (New technology makes people more isolated), 118 (55.4%) and 55 (25.8%) of participants, respectively, either disagreed or neither agreed nor disagreed with the statement. These findings are summarized in Table 2.

Concerning positive attitudes towards technology, in questions 1.1 through 1.6, most participants said to either agree or strongly agree with most statements, resulting in a median score of 4 ± 0.6 (agree). As for negative attitudes towards technology, in questions 1.7 through 1.9, most participants either disagreed or neither agreed nor disagreed with most statements, resulting in a median score of 3 ± 0.8 (neither agree nor disagree). Based on the analysis of the survey results, it can be inferred that a majority of the surveyed population views new technologies as having a positive impact on their daily lives. They believe that new technologies provide them access to information anytime and anywhere and simplify their daily activities. This suggests that globally the sample is receptive to adopting new technology-based solutions, such as the one proposed in this research, confirming its suitability for the current study. It should be noted that the majority of the surveyed sample has a high education level and is composed of young people and young adults, which is consistent with the characteristics of common users of technological solutions [43,44].

Additionally, participants were asked to express their perceptions of safety, security, and comfort in relation to downtown Porto. This is an area of the city that is generally very busy during the day and at night, with many services, restaurants and shops, but also with some narrow, dark and sometimes poorly frequented streets. Questions 2.1. and 2.5. pertain to perceptions of safety, question 2.2. pertains to perceptions of security, and questions 2.3., 2.4., 2.8. and 2.9. pertain to perceptions of comfort. Question 2.6. encompasses all three perceptions and question 2.7. encompasses perceptions of safety and security. This classification was derived from the categorization proposed by [24]. Questions 2.1 through 2.9 addressed a few negative aspects of the area, with a median agreement of 3.0 ± 0.80. In general, most participants consider that it is an area of the city with a lot of cars and people traffic and that they have already felt somehow unsafe walking around there. Individual results for each specific question are presented in Table 3 to facilitate data analysis. These findings allow us to comprehend the factors that affect individuals’ sense of safety, security and comfort while navigating in the city, revealing potential features for new solutions that promote their well-being.

#### 2.1.2. Focus Groups and Interviews

Conducting focus groups and interview sessions provided information about global and individual perceptions of safety and comfort when walking or cycling, considering personal characteristics and specific circumstances of travel. Both participants of the focus group and interviews were selected by convenience and invited to participate. All participants were informed of the details and required to sign an informed consent should they agree to participate. The methodology included a socio-demographic, ethnographic and technology-related questionnaire to perform an ethnographic characterization of the group, followed by an interview based on a semi-structured script with the aim of conducting a categorical/events analysis and designed to explore safety, comfort, and functionality perceptions.

Three focus group sessions with eight participants each and six individual interviews were conducted online using Zoom. Each focus group lasted about 90 min and each interview lasted about 20 min. The duration of the interviews is in line with the ones of related studies, such as [45,46,47], and the number of successful interviews indicates that the population of AMP is open to sharing their personal day-to-day mobility experiences and also that the interviews were well structured. All sessions were recorded and consent was obtained before the recording began. The records were then transcribed, anonymized and subsequently validated by an independent consultant; all records were destroyed after transcription.

First, a questionnaire was applied to the participants of the focus group and individual interviews. The main aim was to make a socio-demographic characterization of this sample. Therefore, the sample included 30 respondents, of which 12 (40.0%) were female, with a mean age of 41.89 ± 15.43 years. Regarding formal education, 23 participants (76.7%) had higher education (BSc or higher). These findings are summarized in Table 4.

Regarding transport modes, most participants traveled on foot (23; 35.9%), with most participants (16; 53.3%) stating no limitations to mobility. However, some statements of limitations to mobility included pregnancy or physical impairments (3; 10%), among others. The prevalence of participants who travel on foot provides insight into their perceptions of using active modes of transportation, which is the main focus of the current study. Additionally, their use of other means of transport enables us to understand their perspective on interactions with VRUs, as well as the emotions that may result from such interactions. These findings are summarized in Table 5.

Technology-wise, most participants (25; 43.1%) used a smartphone mainly for social media (19; 16.7%) and navigation purposes (16; 14.0%). For navigation, Google Maps was the preferred application (21; 48.8%) with the purpose of mainly determining the route to a destination (21; 32.8%) or obtaining the fastest route (17; 26.9%).

Participants reported they had felt unsafe when walking, particularly when alone on night-time journeys, fearing accidents or physical violence. Factors that contribute to the perception of an unsafe route are the absence of people and adequate illumination, lack of surveillance, lack of directions, the environment (e.g., degraded neighborhoods, surroundings with signs of vandalism), intense automobile flow, not enough sidewalks, or sidewalks with obstacles. On the other hand, the comfort notion was reduced due to intense people flow, poor indoor and outdoor air quality, higher ambient noise levels, uneven, damaged, or slippery pavement, inadequate illumination, and steep inclines. Accessibility is affected due to the reduced width of sidewalks, works on public roads, poor placement of urban equipment, and a reduced number of access ramps.

When discussing possible features, participants agreed that knowing about the air quality and noise levels could be interesting, particularly for younger, environmentally conscious generations, people with respiratory conditions, or people traveling by active modes of transport. A feature where ‘unsafe areas’ could be flagged was considered of interest, particularly in countries or areas with higher crime rates, but legal and ethical difficulties of implementation must be thoroughly considered in advance

#### 2.1.3. Comparative Analysis

Since both a familiar feeling as well as an added value are expected by the users, evaluating route planners was a good way to perceive general interactions, and user interface elements, as basic/extended functionality. A total of 9 apps from the Google Play Store were tested, which were categorized as route planners or navigation tools: Just Draw It, Plan My Route, Routin, Zeo, Maplocs, Komoot, Citymapper, Google Maps, and Waze.

Evaluation of the features, visual design, content, and usability of the apps gave a general understanding of their typical structure, highlighting the good examples of what was inspiring and what needed improvement. Examining the bigger picture provided recommendations on how to integrate different elements, drawing inspiration from the successful ideas of other projects.


*Just Draw It and Play My Route*


Strengths: Both share the concept of drawing a route on a map with your finger, featuring the option to snap-in to roads and check route elevation, being generally effective. Just Draw It features a ‘climate change motivator,’ which predicts how much CO_2_ you are cutting down on your carbon footprint by walking and not driving.Weaknesses: The functionality of the routing feature is limited to drawing only on Just Draw It, lacking the option of specifying a specific starting and ending point. This creates difficulties when planning longer routes, as constant manipulation of the screen is required. Additionally, both apps come up short on the capability of editing or adjusting the route. Just Draw It even needs a second app to be able to calculate navigation, and most features are cut off from the free versions.


*Routin and Zeo*


Strengths: These apps primarily function as an organizer by automating the process of determining the optimal route between stops, therefore, saving time that would have been spent on calculating distances. It will consider a few parameters and optimize the route based on all of the stops (for a certain number of credits). All functions are readily available to try before forcing the user to buy extra credits.Weaknesses: Does not contain navigation in itself, as starting an optimized route brings up the Google Maps (or Waze) app with the planned path and associated stopovers.


*Maplocs and Komoot*


Strengths: Both are route planners aimed at hikers and cyclists. Available options feature choosing the type of transportation, perceived fitness level, reverse route and close route loop, live weather and elevation profile throughout the locations. The level of adequacy of the planned route is indicated, based on customization, through parameters such as time, distance, elevation, experience and fitness level required, and additional information such as closures or restrictions, types of roads, and types of surfaces.Weaknesses: Navigation on Maplocs pulls up the Google Maps app or other options depending on preference.


*Citymapper*


Strengths: Called a ‘transit app’ for pedestrians, commuters, and tourists. The strongest suit of this app is having integration with virtually every means of public transportation available. It features rental bicycles, bus, metro, tram, rail, e-scooters, cabs, etc., with real-time scheduling and ticket pricing, presenting to the user several options within a certain window of time and distance. The user profile also features a few stats regarding calorie burn, trees saved, and money saved on each route. Sharing the route brings up a ‘meet me here’ function, which can be sent as a message or e-mail to another user.Weaknesses: Most complaints mention city availability, an outdated interface, or inconsistencies in train/subway schedules. Suggestions refer to public transport’s current capacity.


*Google Maps and Waze*


Strengths: Google Maps supports directions for walking, biking, and even public transportation. Waze is a driver-focused app, being best known for its crowd-sourced approach, as it uses community-driven information about road conditions from its users. Drivers can share real-time data about accidents, traffic incidents, speed limits, speed traps, and other trip information that helps other drivers navigate the fastest possible route. Waze offers real-time info such as road closures, road hazards, traffic alerts and real-time traffic conditions based on driver data. Google Maps has only started to include some of these features recently. Google Maps seems to be more reliable, accurate and has better real-time traffic, but Waze fans praise the app’s functionalities and customization.Weaknesses: Google Maps users think that the app is slower and crashes more, whereas Waze users have problems with Android Auto and GPS connectivity. Maps also only focuses on time and distance, and Waze does not support any other means of transportation besides car and motorcycle.

Route planners often feature the same functionalities with different elements or themes, some offering novel interaction paradigms and some having a particular focus on what the main activity is and who it is directed for. Most applications featured simple interfaces with no noise, with menus usually being simple and clean. Just Draw It, Plan My Route, Routin, Zeo and Maplocs are good examples of the first two parameters since they are navigation tools that are on par with the status quo functionality-wise. Komoot’s focus is on biking, so it adds a layer of relevant information that caters to those specific users; Citymapper’s leverage is based on solid integration with public transport information; Waze is entirely directed at motorized vehicles, relying on a crowdsourcing feature for road event reporting; Google Maps is the go-to for navigation tools, as it is a feature-rich all-rounder regarding functionality and interface guidelines.

In essence, these apps have their strong suits regarding either a specific means of transport and way to move around or a specific functionality. However, as there are no route planners that consider noise pollution or lighting levels as route calculation parameters or take conditioned mobility at any level in a more serious manner, it was possible to glimpse an opportunity that needs to be explored.

### 2.2. Proposed Personalized Route Planner for VRUs

The main goal of this proposal is to give users more control over a route by allowing them to consider their preferences in terms of safety, security, and comfort (see Figure 2).

Raw data can be gathered by a multitude of sensors and cameras placed in the environment of choice, which is relevant to urban living, such as air quality, noise levels, lighting levels, people flow or traffic flow. Other data such as slope levels, accessibility points or mobility schedules and capacity can exist in a knowledge base. Sensors can make decisions within the limitations of embedded processing, reducing traffic on the network and limiting the transfer of sensitive data. Data collected by the sensors is stored in the cloud and processed through data mining, machine learning and deep learning algorithms. The information can then be translated and forwarded to the mobile devices of citizens who have a need for or an interest in it. Management of the service also plays a role in dispatching warnings or alerts regarding certain urban circumstances, such as construction work and blockages, accidents, or weather hazards, among other things.

On a mobile device, the route planner can calculate and present personalized routes, as well as information on the aforementioned parameters from the sensors and knowledge base. The route calculation takes into account the preferences indicated by users in terms of safety, security and comfort parameters and sensor and knowledge-based data. The intersection between these two will allow the route planner to indicate optimized and personalized routes for each individual. The advice provided will be optional and not mandatory, and tasks should be clearly defined, short, and easy to learn to maximize interest and encourage future adoption from potential users.

These users are the ones that are or could be using route planners in cities for daily living, services, commerce, and amenities, as well as for commuting, recreation, and tourism. Areas of interest are envisioned to be hubs for transport/commuting, tourism, living and working, with a considerable afflux of people every day, so users might consult the route planner on different situations, either as part of their daily routine or when special circumstances or needs arise.

Users should be motivated by the opportunity of using a tool that gives them a viable option for a modal shift on how they move around, expecting performance and usage to be as close as possible to the ones broadly used and already on the market. However, users can be quite different, depending on the analysis of parameters of interest, such as sociodemographic factors, preferred means of transport, and special needs—characteristics such as age (gaps), gender, education and culture, language and nationality, residence and workplace, presence of chronic illnesses, mobility issues, vision, hearing, and cognitive impairments. It is important to consider them all, as different approaches might differ according to user type and necessities.

Based on the ambition of aiding VRUs, this concept envisions three modes of transport: walking, biking or scootering, and restricted mobility. The restricted mobility category includes factors such as the use of wheelchairs, crutches, old age or health issues, pregnancy, and baby strollers.

## 3. Illustrative Example of an Innovative Route Planner

The proposed innovative concept of the personalized route planner can be materialized in several practical examples. This section explores an example of a route planner that was developed, representing a proof of concept of the proposed idea. This example was materialized into a functional prototype for mobile devices and extensively tested with potential users throughout four loops of usability testing.

The developed mobile application allows planning a trip using active modes of transport and taking into account personal preferences in terms of safety, security and comfort. Thus, the routes suggested by the application are customized to the users’ preferences to maximize their perception of safety, security and comfort. The routes can be calculated using, for example, an A-Star algorithm, considering individual preferences for each parameter and the real values of that parameter, which allows one to obtain a specific set of weights per person, returning the corresponding route [48]. The application also makes it possible to query information passively, with users being able to access information such as air quality or noise level at a certain point in the city. These and other functionalities of the application are presented below, accompanied by some screenshots of the developed prototype. Finally, this section concludes with the presentation of the tests conducted to evaluate the prototype.

### 3.1. Prototyping

Figma was used to develop successive medium-fidelity screens to build a functional prototype that could later be tested. An initial prototype was developed and subsequent refined ones were based on feedback with each testing iteration.

A short and direct onboarding will be used just on a first-time run, greeting the user and getting them acquainted with the purpose. It will feature a quick explanation of the concept as well as two questions on possible limitations and route parameter preferences (Figure 3). A short section of the overlay tooltip for the function rundown can also be included (Figure 4). Skipping it will assume walking and route parameters by default for each mode of transport. Selecting a condition that could restrict mobility, as well as reordering the route parameters to their preferences, will later affect how a better route is recommended. This can be changed at any time on the profile menu.

The home screen should be familiar to those using route planner/navigation applications, featuring three distinct buttons in the left and right lower corners and the top right corner (Figure 5). The lower right button changes the way the user moves around. Options are walking, bike/scooter, and three types of conditioned mobility which may include old age or health condition, wheelchair usage, and being pregnant/using a baby stroller. Each selection will imply, favor or condition the routes to be presented regarding their associated parameters.

The lower left button allows a user to report a selection of temporary events that may be of interest to other users. Users can pinpoint places with unusual pedestrian density, accidents, blockages or roadworks, damaged pavement, and issues with urban furniture and mark accessibility points for those who need them most (as they are missing from most route planners). They can associate a picture or a small description with their report, which, when submitted, will be available in a specific section for other users to consult (Figure 6).

The upper right button lets the user access a selection of real-time information from processed sensor data on interactive map layers (Figure 7 and Figure 8). These sensor stations can be scattered around the map in specific spots and rely on different systems such as cameras, air quality sensors, light sensors, or microphones to capture data around them. The environment layer provides information regarding air quality, noise and lighting levels (Figure 7); the people layer provides information about current pedestrian density; the accessibility layer allows one to check for accessibility spots, such as ramps, accessible bathrooms, elevators and escalators; the mobility layer shows e-scooter and e-bike pickup points as well as how many are left, bus, train and subway stations, as well as brief information about the line or oncoming departing vehicles; the traffic and slope layers allow the user to check current road traffic, as well as the average slope level on a certain area; the alerts layer brings out all the available crowdsourced reports both from other users, as well as the city administration for others to consult. There is also the option to approve or disapprove the reports at hand.

The lower navigation bar gives access to the (home/current) navigation menu, the profile section, and another one with previously saved or taken routes, as well as predefined quick addresses (Figure 9).

Inside the profile menu, there are several tabs (Figure 10). The stats tab features (see Figure 9 above), as the name states, various statistics about general usage, along with environmentally themed gamification components (calories spent by walking or cycling, trees saved/CO_2_ not emitted by not driving, money saved by not driving or using public transport services, etc.). The preferences tab allows the user to make a few changes regarding layout and customization. In the route aspects tab, users can set the default profile regarding route parameters to their own default when they see fit. Karma points are a gamification implementation that encourages users to accurately provide useful reports for others, as well as interacting with the ones posted by the community. The collected points can, at times, be exchanged for rewards that promote good behavior and encourage active mobility.

Not forgetting the main functionality, which is still expanded route planning with personalized results, users can simply choose a destination and will be presented with three routes (Figure 11 and Figure 12). A recommended route will be based on the route parameters which they reordered to their tastes, as well as on their current means of movement. Two other routes can also be selected. They always feature a trade-off that presents the user with a choice to experiment with different routes based on current mood or need. There is also an option to expand in detail the different aspects that make up the chosen route. Starting a route initiates the navigation screen, which provides the options to cancel the route and report an event, as well as an SOS function for night-time travel. This allows the users to access a menu where they can quickly call the national emergency number, as well as calling or sharing their location with their appointed emergency contacts. Ending a route will trigger a screen where the users will be asked to provide some feedback about their journey. Answering it will aid the proper automatic tuning of the user’s real preferences and, subsequently, the recommendation of future routes.

### 3.2. Prototype Evaluation

Although users were expected to possess a satisfactory level of ICT usage skills and a positive attitude towards technology, there was still a need to assess an interface that facilitated their ability to complete typical predicted tasks in a satisfactory manner. The main aim was to analyze the usability of the design, information flow, and information architecture by resourcing to the collection of both objective and subjective measures. Four loops of usability tests were set to promote multiple moments for user participation and feedback, as well as for problem correction. It assessed the proposal’s look and feel appeal to a superficial degree and analyzed to what extent the prototype’s organization made it easy to find the information and where it was contained while keeping track of where on it the user was. We also evaluated how pleasant, satisfying and interesting was the user interaction with the prototype (e.g., perceptions on how easy it was to use it; if participants would be willing to use it in the future); The test covered several functional features, providing an ‘all-rounder’ look of what should do and look like. The next sections describe the evaluation process that was followed as well as the main results achieved.

#### 3.2.1. Sessions

Four evaluation loops were conducted. Testing sessions ranged from approximately 25 to 45 min, depending on the current loop, and were carried out in a presential setting. Participants were handed a smartphone and asked to test the prototype by engaging with the interface (Figure 13). The sessions were designed to capture data on the participant’s navigational choices, task completion rates, overall satisfaction, and additional feedback. Participants were instructed to perform tasks and think aloud while doing so. Their navigational choices, vocalizations, and any relevant non-verbal behavior were recorded, as were their thoughts, actions, doubts, and questions. The evaluator also provided their own observation at times.

Not all sessions were structured the same way. The first and second loops featured a short discussion about what the perceived purposed and perceived audience were, four prototype tasks and a post-test questionnaire. The third and fourth loops, performed at a later stage of prototype maturity, were performed to further polish the proposed solution, featuring a short introductory discussion and a supervised free roaming while thinking aloud, with no specific tasks, while also featuring the post-test questionnaire.

#### 3.2.2. Participants

The concern in recruiting heterogeneous participants regarding demographics and ICT skills was present, but due to the nature of testing being on a tight schedule and at a time when participants had less availability for face-to-face testing, quotas regarding those parameters had to be less strict to include a larger number of users. They were recruited at both the Faculty of Engineering and the Faculty of Sciences of the University of Porto, resorting to snowball sampling. All information collected from users was gathered exclusively for testing and improving the prototype. All participants at the time of recruitment were provided with information and a description of the study’s aims. They were free to withdraw from the project and request the deletion of their data at any time without the need for justification or incurring penalties.

The first loop featured 19 users, the second 18, the third 6, and the fourth 4, bringing the total to 47 performed tests. Regarding the participants’ demographics and characteristics, it was possible to gather a heterogeneous group, where from the total 47, 21 were female and 26 were male. Participating individuals were students or teachers at the institutions, which resulted in a sample with a higher education level and above-average ICT usage skills. In total, 8 had a bachelor’s degree, 26 had a master’s degree and 13 had a doctorate degree. Age-wise, 12 were in the 20–24 age group, 8 in the 25–29 age group, 13 in the 30–34 age group, 9 in the 35–39 age group, 3 in the 40–44 age group, 1 in the 45–49 age group and 1 in the 50+ age group.

#### 3.2.3. Tasks

Participants proposed a set of tasks without contextualization with hypothetical scenarios. A few prompts from the evaluators were acceptable to contextualize the user on the objectives and proposed functionalities.

The tasks regarding the use of key features of the prototype were:Start a route while going on foot;Check the extended route aspects;Check the air quality status on a certain area of the map;Change your personal route choice parameters.

They were evaluated as ‘not completed’, ‘completed with difficulty of acceptable’ prompts (counting as a partial success) or ‘easily completed’ (counting as a full success). The stop criterion was applied when one of the three conditions occurred: a. the users completed the task successfully; b. the users said they completed the task, even if they did not; or c. the users wanted to or decided to give up.

A high rate of success was expected due to tasks being relatively simple and straightforward, providing the participants with the opportunity to explore with few constrictions and a clear goal. We confirmed our expectations (Table 6) while still receiving valuable feedback from the users while they were performing those tasks. No task was left incomplete, so it was only a matter of checking paths that would not work as well. Tasks with a higher percentage of partial successes meant that the UI element that was attached to it probably was not very perceptible and had to be redesigned.

#### 3.2.4. Retrospective Questionnaire

The questionnaire consisted of open questions that were used to understand and gather the participant’s feedback and key features of the prototype, color schemes, layout, language used, readability, perceived usefulness, and perceived ease of use. These would also include overall impressions, missing features they would expect to see, organization, etc. Participants were asked the following questions:What did you like the most?What did you not like as much?How was the layout (organization)? Content (easy to read, general spacing)? Colors (soft, flashy)? Language (easy to understand)?Anything you were expecting to see and did not?

The test administrators also gathered subjective measures of participants’ satisfaction with the platform by handing them an adaptation of the System Usability Scale [49]. Participants had to position themselves on a five-point Likert scale ranging from “Strongly Disagree” to “Strongly Agree” regarding a few statements (see Table 7).

The percentage of the agreement represents a combination of ‘Agree’ (4) and ‘Strongly Agree’ (5) ratings. Agreement on statements can be understood as good indicators of the app’s usability, except for questions 2 and 5 (inverted questions), where agreement on statements can be understood as bad indicators of the platform’s usability.

Among other questions, 85% of the participants agreed that the app was well organized and easy to use, with only 21% admitting they would need support to be able to use it. The vast majority deemed the app as useful, with 77% saying that they would like to use the app frequently. Most people did not have trouble keeping track of where they were in the app, with only 11% stating they felt lost at times. However, participants want, inherently, to be nice and not a burden, a few exceptions aside (social desirability bias). It is known that sometimes, the users’ rating does not exactly translate to what they really think, so these opinions should be taken with a grain of salt. Thus, to complement these ratings, a qualitative synthesis of the contents provided by the participants is presented in the following section.

#### 3.2.5. Subjective Evaluation

Participants were then invited to subjectively evaluate the aspects of the app they consider strengths and the elements they consider to be limitations, as well as mentioning absent elements they expected to be present and whether they would consider future usage.

Regarding layout aspects, such as organization, content, colors and language, most participants (37 out of 47) had a positive response to the layout, viewing it as well-organized and clear, with a suitable tone and important content, commending the color scheme and visual design.

“Pleasant design, good readability, not too cluttered, nice color scheme.” (U1)“Looks simple, neat, and tidy… content easy to read.” (U7)“Pleasant layout, good use of the color palette, accessible language.” (U18)“… layout and colors are good… you can’t stray away from being close to Google Maps.” (U24)“Everything looks appropriate, I like the content… everything is clear and clean.” (U33)

Five participants had negative perceptions of the layout, viewing the interface as complex, confusing, or not to their liking. Five others had nothing relevant to say.

“I don’t really like the way you present the route filters.” (U2)“At some places, the text seems too small and not all the icons are intuitive.” (U9)“I don’t see where I’m able to press things at all times… simplify things a bit where you can.” (U27)“… layout is too similar to Google Maps, I would like to see something different.” (U29)“I don’t really read that much text, I’m more of a visual person, so I prefer the message to be passed on by images, icons, things like that.” (U42)

When it came to the most liked features, participants leaned on the novelty of having different parameters to choose from on route planning (32 out of 47), with 15 vocalizations about the relevance of consulting information on-the-fly, regarding those parameters. Twelve participants found the gamification and reward system (karma points) very interesting and a differentiating factor.

“… saved trips, profile stats, route parameters, all great things that put this app apart from the others… the lighting level information is really important, especially at night.” (U5)“… I understand the importance of air quality for people who are allergic or suffer from it. I can relate to the other parameters.” (U17)“… having more control over the route parameters besides time and distance, and considering different types of ‘movement’… this filter/layer information is very unique…” (U23)“… love these Karma points… would definitely make me try the app just to see what would happen with the rewards.” (U31)“… I never thought about the difference between ‘just walking’ and walking with a baby stroller when I needed it. I know how different those routes can be.” (U45)

On the other hand, participants that pointed out issues (23 out of 47) focused on functions that were not as intuitive, a few oversights and a couple of expected tweaks discovered during testing.

“… way too difficult to access the route details, it needs to be more intuitive.” (U11)“I don’t believe people would use the crowdsourcing/reporting feature that much.” (U27)“… should include some kind of intro to the app on the onboarding… it is confusing because there is no context of anything… I know because you explained it to me, but if you hadn’t…” (U32)“… struggled a bit with this drag and reorder thing on the route parameters…” (U35)“… not really a feature but maybe an oversight from your end… because I think you should support changing those preferences on-the-fly instead of having to commit to what you set up earlier.” (U42)

When asked about expected functionality or features that were not present, a few users mentioned a variety of expectations or shared personal ideas, including suggested additional features. Some were considered and added to the final prototype. The majority (30 out of 47) had nothing relevant to add.

“… would definitely add a crowdsourcing function to report accidents, events, etc., around town… like Waze.” (U9)“… integration with public transport, but I don’t know if that’s what you’re aiming for here.” (U19)“… voice navigation and points of interest… could be great for tourists or newcomers to a town that plan to work there or just stay for longer.” (U27)“… a higher degree of customization of the saved routes.” (U35)“… better with dog-friendly routes or exercise-friendly routes…” (U41)

All users had something to say regarding possible usage, with most of the answers falling on the theme of ‘added value’ over other apps and trying it ‘at least once’ if it provided reliable results.

“If I had mobility restrictions, it would be interesting. The added value beyond what Google Maps offers is also great.” (U4)“If I lived in a place that I didn’t know that well and wanted to explore everything, of course.” (U10)“If it does all that Google Maps does and then some, I will use it… being able to check the passive information and real data of urban monitoring would be great.” (U15)“… could try it for a while… if it were reliable and gave me the expected results when calculating the routes, I would continue to use it.” (U18)“… there are two distinguishable features that could make me use this app: the focus on conditioned mobility in several ways, and being able to tailor the route to your liking.” (U36)

These tests revealed the perceived value of the proposed solution, particularly in the integration of multiple parameters in route calculation and the ability to account for various user characteristics and constraints. This was deemed a crucial consideration in promoting the selection of more sustainable modes of transport.

## 4. Discussion

This study focuses on the importance of providing meaningful information to VRUs to support their perceptions of personal safety, security and comfort while moving within urban spaces. This is achieved through the proposal of a new concept of route planning, which is hinged on the potential of using urban sensor data to increase parameter options when presenting a route. Current route planners mostly focus on two dimensions, namely time and distance, which might come up short when supporting personal perceptions, as people might need and want different things when going from point A to point B. Considering new aspects and different dimensions, such as air quality, noise levels, accessibility, or people density, opens the possibility for route customization to a higher degree, which can be a tool to persuade citizens to shift to more sustainable means of transport.

After an extensive data collection process consisting of questionnaires, focus group sessions and interviews, it was found that, although people are familiar with route planners, these only satisfy their basic mobility needs, not being adaptable to their personal concerns and preferences or taking into account determining factors in their travel experiences, such as safety, security and comfort. For example, participants stated that they already felt somehow unsafe or uncomfortable while walking or cycling, pointing out several causative factors such as the absence of people and adequate lighting in the streets, lack of surveillance, the intense flow of cars, lack of exclusive lanes for pedestrians and cyclists, poor outdoor air quality, high noise levels, and uneven, damaged or slippery pavements. It was also evident that these perceptions vary from person to person, making it necessary to provide personalized information for each individual. It is important to exercise caution when interpreting these results, as the number of respondents, despite efforts to widely distribute the survey, was not sufficient to achieve a higher level of statistical significance. Additionally, the survey was administered to the population of the Metropolitan Area of Porto, Portugal, and it would be beneficial to replicate this study in other metropolitan areas and cities to generalize the findings.

That said, there was a need to materialize the whole concept into a concrete prototype that could be subjected to user testing and feedback in a way that the proposal could become tangible for potential users. The testing sessions provided insights not only into personal preferences but also into interface and core functionality suggestions. Throughout all four prototype iterations, a substantial majority of participants were able to fully complete the proposed tasks that were associated with the key functions, which was a reassurance of the good level of usability provided. General impressions regarding the visual aspect were favorable, regarding its layout, color and content as very appealing. The subjective evaluation and verbal feedback yielded the most encouraging result: the recognition of the usefulness of the concept, which ultimately validates it. Participants had a satisfying experience using the prototype, largely due to the recognition of added value to a product they already know and use.

## 5. Conclusions

Changing paradigms is not an easy task. However, an effort to promote a modal shift to the use of more sustainable transport means is imperative. As being conscious about creating a better environment and improving health should be the core concern, having tools that support that thought will help this endeavor. This study showed that there is an opportunity to improve on well-known and well-used tools, such as route planners, to support different types of users, not only in terms of mobility restrictions but also by paying attention to their specific objectives and personal perceptions of safety, security, comfort, or accessibility. Providing a higher degree of power and customization to users gives them a sense of better controlling their current moods, goals or concerns, which in this case, could in turn, make them feel safer and more comfortable using tools that promote active modes of transport, leading to an increase in their use. The aim was not to compete with the mapping giants already on the market but, more importantly, to highlight the value of considering new aspects in route planners.

Future research will involve testing the proposed mobile application on a larger scale and in a real-world environment to determine its effectiveness in a real-world setting and its impact on people’s daily lives. From the perspective of implementing the proposed solution, it would be valuable to investigate the multicriteria optimization problem presented in this article, taking into account not only individual optimization but also the relationship and implications with the interests of various user groups.

## Figures and Tables

**Figure 1 ijerph-20-03027-f001:**
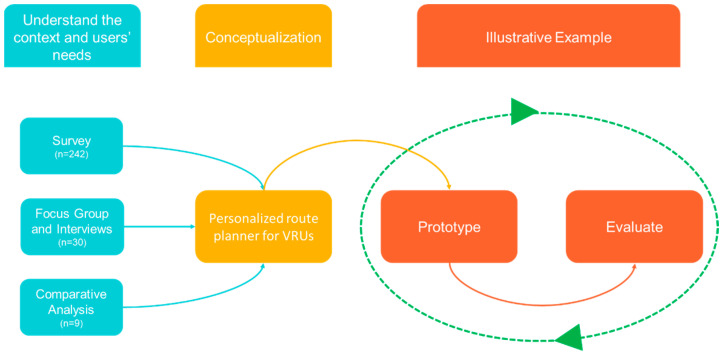
Methodological approach followed to develop the proposed route planner concept.

**Figure 2 ijerph-20-03027-f002:**
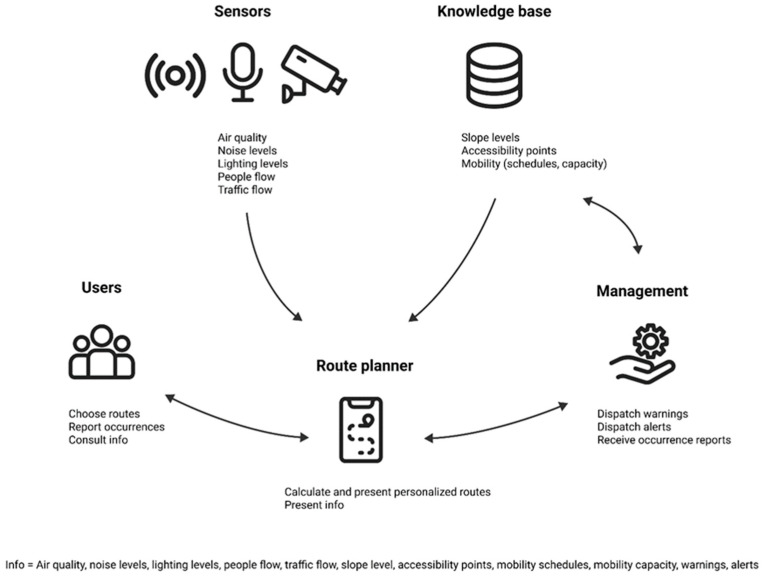
Components of the newly proposed route planner concept.

**Figure 3 ijerph-20-03027-f003:**
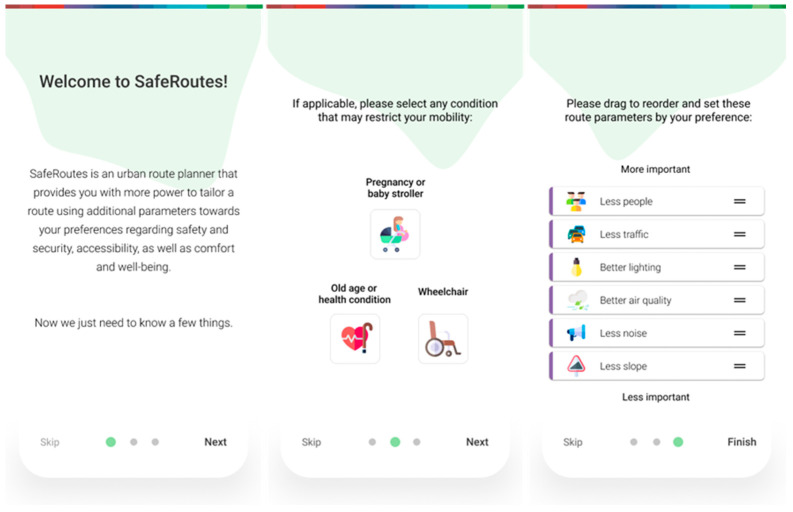
Screens for the introduction to the route planner with onboarding questions.

**Figure 4 ijerph-20-03027-f004:**
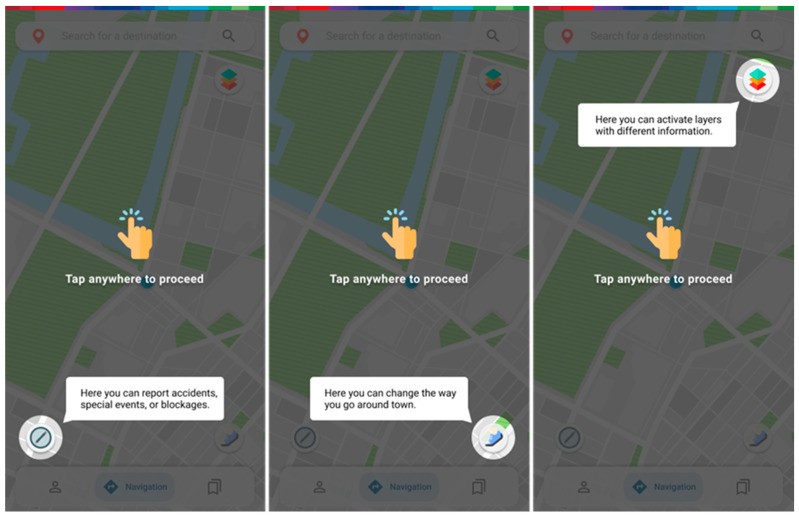
Screens for the explanation of basic functionalities.

**Figure 5 ijerph-20-03027-f005:**
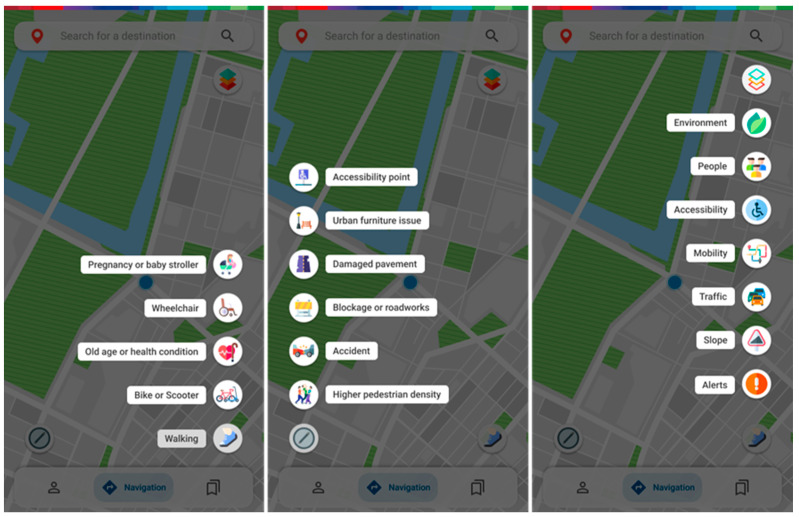
Screens for choosing the mode of transport, reporting issues or events, and consulting information.

**Figure 6 ijerph-20-03027-f006:**
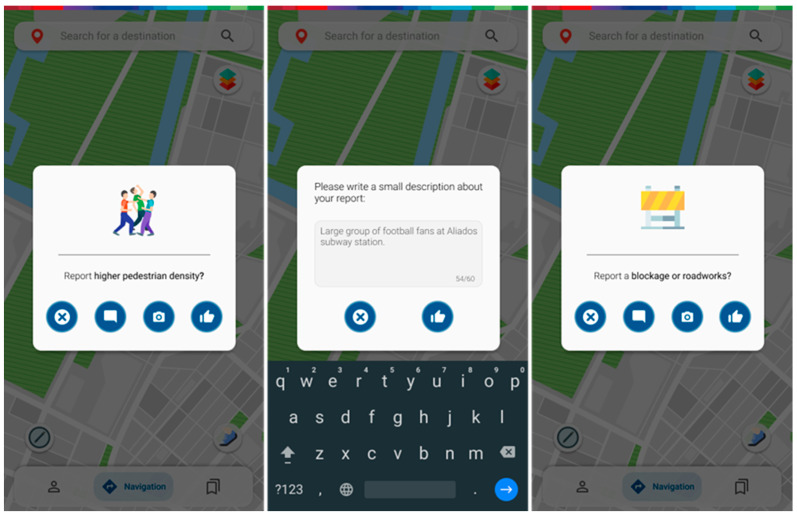
Issue reporting screens.

**Figure 7 ijerph-20-03027-f007:**
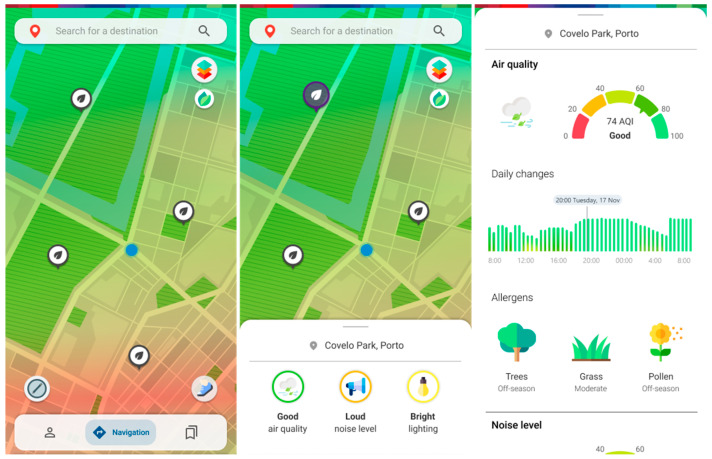
Screens for air quality, noise level and lighting level section, with heatmap and further information.

**Figure 8 ijerph-20-03027-f008:**
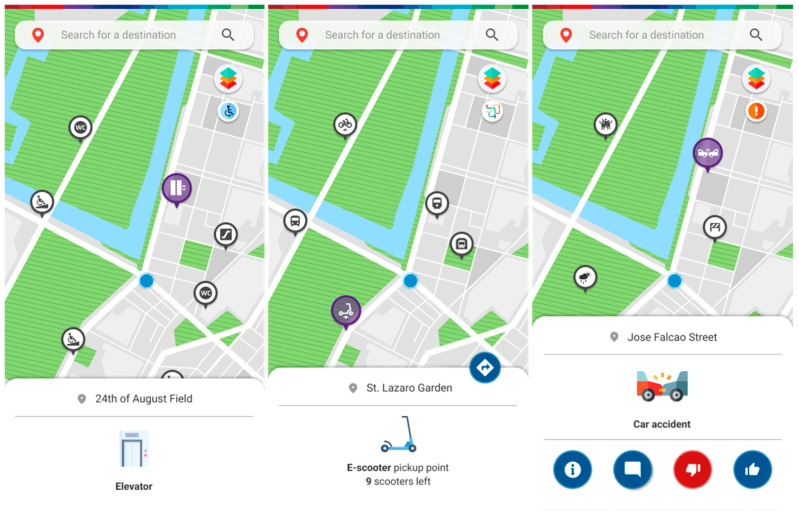
Screens depicting information on the location of elevators, scooter pickup points, and a car accident site.

**Figure 9 ijerph-20-03027-f009:**
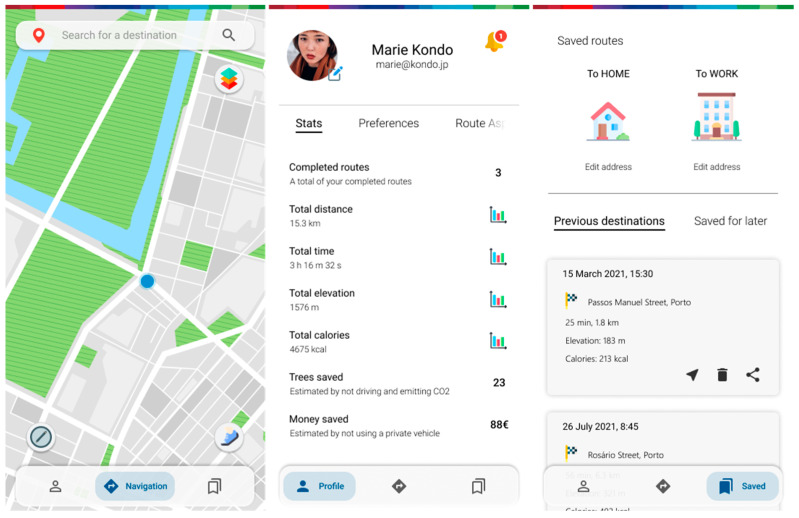
Home screen, profile section, and previously saved locations section.

**Figure 10 ijerph-20-03027-f010:**
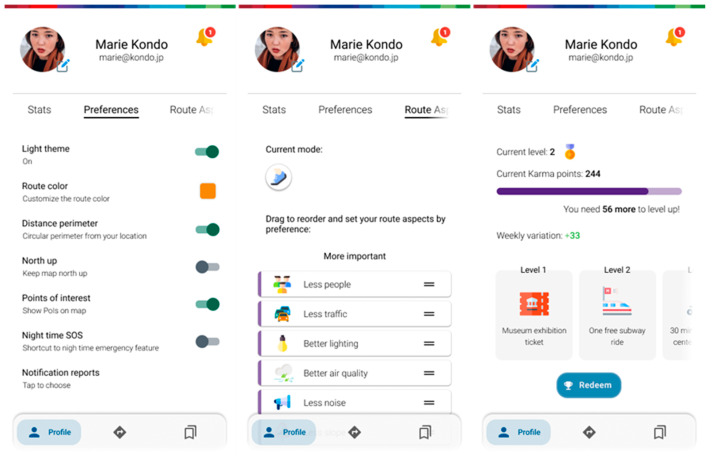
Screens of the profile section featuring user preferences, ordering route aspect importance, and a rewards section.

**Figure 11 ijerph-20-03027-f011:**
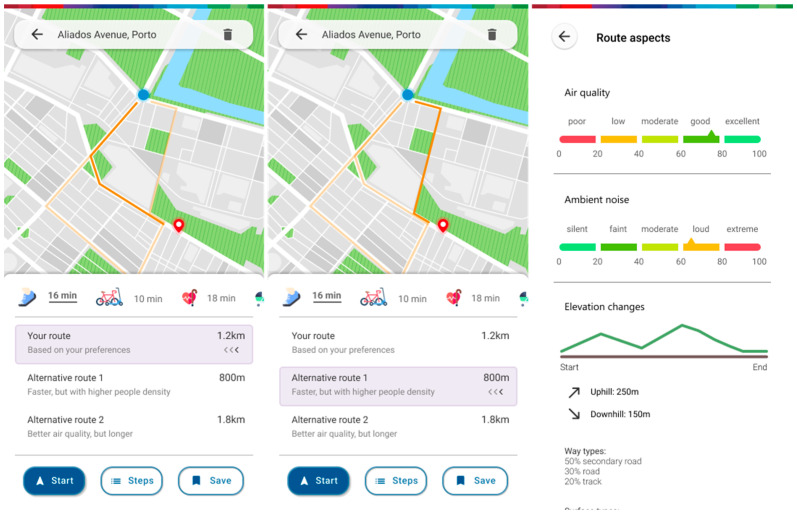
Navigation screens featuring a preferred route, choice of an alternative route, and further information regarding route aspects of a certain route.

**Figure 12 ijerph-20-03027-f012:**
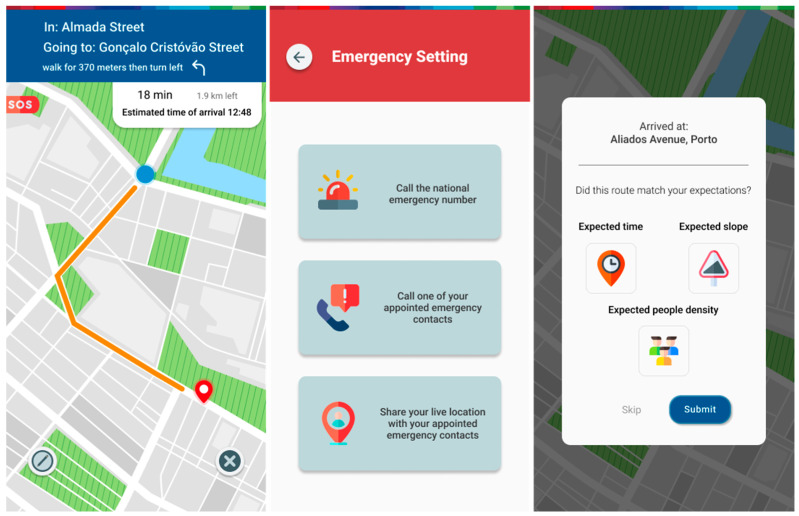
Route ongoing navigation screen, SOS screen, route finished feedback screen.

**Figure 13 ijerph-20-03027-f013:**
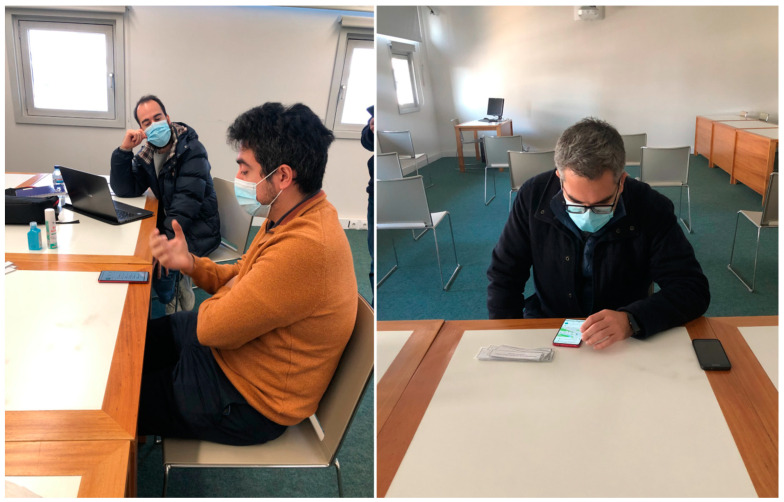
Participants testing the interface.

**Table 1 ijerph-20-03027-t001:** Socio-demographic characterization of the surveyed population.

*N* = 242		*f* (%)
Gender *n* = 240	Male	103 (42.9)
	Female	137 (57.1)
Age *n* = 241	20–29	32 (13.3)
	30–39	64 (26.6)
	40–49	65 (27.0)
	50–59	52 (21.6)
	60+	28 (11.6)
Formal education *n* = 237	ISCED 2: Lower secondary education	1 (0.4)
	ISCED 3: Upper secondary education	14 (5.9)
	ISCED 5: Short-cycle tertiary education	1 (0.4)
	ISCED 6: Bachelor’s or equivalent level	198 (83.5)
	ISCED 7: Master’s or equivalent level	3 (1.3)
	ISCED 8: Doctoral or equivalent level	20 (8.4)

**Table 2 ijerph-20-03027-t002:** Attitudes towards the technology of the surveyed population.

*N* = 242		Strongly Disagree	Disagree	Neither Agree nor Disagree	Agree	Strongly Agree
						*f* (%)
Q1.1 *n* = 212	I feel it is important to be able to find any information whenever I want online.	3 (1.4)	5 (2.4)	18 (26.8)	97 (45.8)	89 (42.0)
Q1.2 *n* = 213	I feel it is important to be able to access the Internet any time I want.	13(6.1)	40 (18.8)	71 (33.3)	75 (35.2)	14 (6.6)
Q1.3 *n* = 213	I think it is important to keep up with the latest trends in technology.	1 (0.5)	4 (1.9)	9 (4.2)	90 (42.3)	109 (51.2)
Q1.4 *n* = 207	Technology will provide solutions to many of our problems.	3 (1.4)	17 (8.2)	67 (32.4)	93 (44.9)	27 (13.0)
Q1.5 *n* = 213	With technology anything is possible.	0 (0.0)	1 (0.5)	27 (12.7)	130 (61.0)	55 (25.8)
Q1.6 *n* = 212	I feel that I get more accomplished because of technology.	26 (12.3)	70 (33.0)	74 (34.9)	34 (16.0)	8 (3.8)
Q1.7 *n* = 212	New technology makes people waste too much time.	18 (8.5)	60 (28.3)	72 (34.0)	48 (22.6)	14 (6.6)
Q1.8 *n* = 213	New technology makes life more complicated.	14 (6.6)	58 (27.2)	56 (26.3)	71 (33.3)	14 (6.6)
Q1.9 *n* = 213	New technology makes people more isolated	23 (10.8)	118 (55.4)	55 (25.8)	16 (7.5)	1 (0.5)

**Table 3 ijerph-20-03027-t003:** Surveyed population agreement ratios about positive and negative area aspects.

*N* = 242		Strongly Disagree	Disagree	Neither Agree nor Disagree	Agree	Strongly Agree
						*f* (%)
Q2.1 *n* = 213	I have felt unsafe when walking in this area.	7 (3.3)	45 (21.1)	57 (26.8)	88 (41.3)	16 (7.5)
Q2.2 *n* = 214	I already got lost while traveling in this area.	22(10.3)	84 (39.3)	30 (14.0)	71 (33.2)	7 (3.3)
Q2.3 *n* = 212	This area is too noisy.	67 (31.6)	107 (50.5)	15 (7.1)	21 (9.9)	2 (0.9)
Q2.4 *n* = 212	During an event/concert, I prefer to move away to a place more distant from the stage.	6 (2.8)	56 (26.4)	55 (25.9)	84 (39.6)	11 (5.2)
Q2.5 *n* = 212	This area has too much car traffic.	4 (1.9)	44 (20.8)	52 (24.5)	96 (45.3)	16 (7.5)
Q2.6 *n* = 214	This area has too much pedestrian traffic.	3 (1.4)	19 (8.9)	42 (19.6)	111 (51.9)	39 (18.2)
Q2.7 *n* = 212	This area has enough pedestrian zones.	22 (10.4)	74 (34.9)	60 (28.3)	44 (20.8)	12 (5.7)
Q2.8 *n* = 212	This area is difficult to access for people with mobility issues.	8 (3.8)	82 (38.7)	57 (26.9)	61 (28.8)	4 (1.9)
Q2.9 *n* = 212	It is difficult to find parking/parking spaces in this area	0 (0.0)	8 (3.8)	53 (25.0)	115 (54.2)	36 (17.0)

**Table 4 ijerph-20-03027-t004:** Socio-demographic characterization of the focus groups and individual interviews population.

*N* = 30		*f* (%)
Gender	Male	18 (60.0)
	Female	12 (40.0)
Age	Mean ± SD	41.89 ± 15.43
	Min; Max	19; 73
Formal education	Higher education (BSc)	15 (50.0)
	Postgraduate studies (MSc or PhD)	8 (26.7)
	Secondary school	7 (23.3)

**Table 5 ijerph-20-03027-t005:** Transport mode preference and limitations to the mobility of the focus groups and individual interviews population.

*N* = 30		*f* (%)
Transport mode *n* = 64	On foot	23 (35.9)
	Bus	3 (4.7)
	Underground	6 (9.4)
	Train	2 (3.1)
	Car	22 (34.4)
	Transport Applications	2 (3.1)
	Bicycle	3 (4.7)
	Wheelchair	2 (3.1)
	Car Sharing	1 (1.6
Limitations to mobility *n* = 30	None	16 (53.3)
	Baby cart	2 (6.7)
	Pregnancy	3 (10.0)
	Wheelchair	2 (6.7)
	Respiratory Disease	2 (6.7)
	Physical Impairment	3 (10.0)
	Bone, Joint or Muscle Impairment	2 (6.7)

**Table 6 ijerph-20-03027-t006:** Task completion results from the first two usability loops.

Task #	Task Name	Partial Success	Full Success
1	Start a route while going on foot	0%	100%
2	Check the extended route aspects	22%	78%
3	Check the air quality status on a certain area of the map	8%	92%
4	Change your personal route choice parameters	27%	73%

**Table 7 ijerph-20-03027-t007:** SUS agreement rates from the four usability loops.

Question	Agreement (%)
1. I think the prototype was easy to use.	85
2. I think that I would need support to be able to use this prototype.	21
3. I think the prototype was well organized.	85
4. I could get the information quickly.	96
5. I found it difficult to keep track of where I was.	11
6. I think that most people would learn to use this very quickly.	85
7. I think that I would like to use this frequently.	77
8. I think the concept is useful.	85
9. I found the prototype pleasant to use.	89

## Data Availability

Data is unavailable due to privacy restrictions.

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
