# Peer review of "A New Perspective on Supporting Vulnerable Road Users’ Safety, Security and Comfort through Personalized Route Planning"

_ijerph, 2023, doi:10.3390/ijerph20043027_

Round 1

Reviewer 1 Report

The strength of the present manuscript lies in the use of various analytic methods for exploring the opinions and perceptions of vulnerable road users. I also appreciate the clarity with which the authors illustrate the results of their research and proposed app for route planning. Having said this, I do have some comments that I hope the authors could address.

1.    The manuscript is mainly composed of two parts: a survey of VRUs’ needs and concerns, and the design of route planning app. However, the connection between these two parts is not clearly presented. For examples, it’s not clear whether or not the survey participants’ attitudes towards technology (Table 2) and perception of the study area (Table 3) are important factors influencing the design of the route planning app.  

2.    Although in the section of introduction the authors have emphasized the importance of promoting a modal shift to the use of a of more sustainable transport means, it’s a pity that not many of the questions in the survey touched upon the issue. Especially in the section of app evaluation, there should be more discussion about the participants’ feedback on the choice of transport means.

3.    The content of 2.1.3 Comparative Analysis should be a valuable reference for the authors’ design of route planning app, but right now the comparison is too simplified. The authors may provide more details regarding the design of the related apps and further specify their similarities and differences, strengths and weaknesses.

Reviewer 2 Report

-          In your abstract the case study was not clear.

-          Why 242 respondents are enough? Justify it.

-          Is the frequency of each category of respondents in accordance with your population distribution?

-          Line 166. You mean access to internet when cycling for example? Is it safe?

-          I cannot understand the relationship between questions in Table 2 and your research objectives? For example the last question relates to the technology and isolation. Please explain about the relationships more.

-          Some of the questions in Table 2 are too general to answer. For example “with technology anything is possible”. This anything is too broad.

-          The answers to questions in Table 2 depend to age, income, and education and among others. For example young people are more involved with new technologies than old people. Thus the attitude toward technology depends on your sample. How we can verify the reliability of the outputs then?

-          Please outline which questions of table 3 relate to safety, which one to security and which one to comfort? I could not specify which one relates to security for example.

-          It is really hard to have an interview of about 20 minutes with people nowadays. I am really eager to know how you encouraged them for this interview.

-          I think the mode that people now use frequently for their daily commutes as outlined in table 5, affect their answers. How you justify this?

-          When reaching to section 2.1.3 I became completely confused about your research method. Please provide a flowchart or a section in the method to give us a complete road map and then present the details.

-          Based on lines 272-273 you are trying to facilitate the condition for each person. However, it might contradict with the total benefits of the network and also the interests of a group of people might lead to costs for another group of people.

-          I could not understand how your model works based on section 2.2. The overall idea and data collection was clear. But how you did it is not clear.

-          Lines 317 to 320. How you do this procedure?

-          I think the main drawback of the paper returns to presentation of the method and calculations. I could not totally convinced how your method add safety, security and comfort to route finding. It seems an optimization problem.

Round 2

Reviewer 2 Report

1-      In lines 156-163, you have tried to justify the sample size. How you reached to 208 is still vague. Based on what reference or formulation you have calculated it? I have never seen 85 percent confidence in any research paper.

2-    - “Is the frequency of each category of respondents in accordance with your population distribution? Authors: This has been made clear in section 2.1.1.” I could not see any justification about this comment.

3-    “I cannot understand the relationship between questions in Table 2 and your research objectives? For example the last question relates to the technology and isolation. Please explain about the relationships more.” I was not convinced by the response to this comment.

4-    “Some of the questions in Table 2 are too general to answer. For example “with technology anything is possible”. This anything is too broad.” I was not convinced by the response to this comment.

5-    “The answers to questions in Table 2 depend to age, income, and education and among others. For example young people are more involved with new technologies than old people. Thus the attitude toward technology depends on your sample. How we can verify the reliability of the outputs then?” I was not convinced by the response to this comment..

6-    - Please outline which questions of table 3 relate to safety, which one to security and which one to comfort? I could not specify which one relates to security for example.” In which lines?

7-    It is really hard to have an interview of about 20 minutes with people nowadays. I am really eager to know how you encouraged them for this interview.” The answer to this question did not relate to my question.

8-    “I think the mode that people now use frequently for their daily commutes as outlined in table 5, affect their answers. How you justify this?” Which lines?

9-    Refer to lines instead of sections in the response letter.

10- Your responses to most of my comments were not enough or convincing.
